Lack of genetic structure in greylag goose (Anser anser) populations along the European Atlantic flyway

Pellegrino Irene 1
Cucco Marco 1 cucco@unipmn.it
Follestad Arne 2
Boos Mathieu 3
1 University of Piemonte Orientale, DISIT , Alessandria , Italy
2 Norsk Institutt for Naturfoskning , Trondheim , Norway
3 Naturaconst@, Research Agency in Applied Ecology , Wilshausen , France
Wink Michael
Electronic publication date: 2015 Aug 13
Publication date: 2015
Volume: 3
Electronic Location ID: e1161
Received 2015 Apr 7; Accepted 2015 Jul 17
Copyright: © 2015 Pellegrino et al.
Copyright year: 2015
Copyright holder: Pellegrino et al.
License: This is an open access article distributed under the terms of the Creative Commons Attribution License, which permits unrestricted use, distribution, reproduction and adaptation in any medium and for any purpose provided that it is properly attributed. For attribution, the original author(s), title, publication source (PeerJ) and either DOI or URL of the article must be cited.
License URL: https://creativecommons.org/licenses/by/4.0/

Keywords: mtDNA, Microsatellites, Greylag goose, Genetic structure, France and Norway

Funding: French National Fund The study was supported by the French National Fund for Biological Research in Wildlife Species to Mathieu Boos. The funders had no role in study design, data collection and analysis, decision to publish, or preparation of the manuscript.

==============================
Greylag goose populations are steadily increasing in north-western Europe. Although individuals breeding in the Netherlands have been considered mainly sedentary birds, those from Scandinavia or northern Germany fly towards their winter quarters, namely over France as far as Spain. This study aimed to determine the genetic structure of these birds, and to evaluate how goose populations mix. We used mitochondrial DNA and microsatellites from individuals distributed throughout the European Atlantic flyway, from breeding sites in Norway and the Netherlands to stopover and wintering sites in northern and south-western France. The mtDNA marker (CR1 D-Loop, 288 bp sequence, 144 ind.) showed 23 different haplotypes. The genetic distances amongst individuals sampled in Norway, northern France and the Netherlands were low (range 0.012–0.013). Individuals in south-western France showed a slightly higher genetic distance compared to all other sampling areas (ranges 0.018–0.022). The NJ tree does not show evidence of any single clades grouping together all individuals from the same geographic area. Besides, individuals from each site are found in different branches. Bayesian clustering procedures on 14 microsatellites (169 individuals) did not detect any geographically distinct cluster, and a high genetic admixture was recorded in all studied areas except for the individuals from the breeding sites in Norway, which were genetically very close. Estimation of migration rates through Bayesian inference confirms the scenario for the current mixing of goose populations.

Introduction

The greylag goose (Anser anser) is widespread throughout the Palearctic. In Europe, the main breeding populations are located in central and northern countries, and the species rarely breeds in Mediterranean areas (Cramp, 1977; Hagemeijer & Blair, 1997; BirdLife International, 2004). European populations show different patterns of movement. Although individuals breeding in Scotland and the Netherlands are considered sedentary birds (Delany & Scott, 2006), those from Scandinavia or central Europe fly longer distances, namely over France to Spain, with some individuals reaching north Africa (Fox et al., 2010; Nilsson et al., 2013). Icelandic breeders winter in Ireland and Britain, and greylags from Russia reach the regions bordering the eastern Mediterranean, Black and Caspian seas. Individuals with morphological characters ascribed to the oriental subspecies rubrirostris have been observed on rare occasions in western Europe (Cramp, 1977).

Widescale movement patterns have been studied through the recapture or resighting of marked birds (coloured neck collars and leg rings, see Nilsson et al., 2013). These methods gave valuable information about the origin of birds that were found in moulting areas (Nilsson, Kahlert & Persson, 2001), flying, or staging in winter quarters. Birds from Sweden and Norway fly to Denmark and/or the Netherlands (SOVON, 1987; Persson, 1993; Andersson et al., 2001). Recent monitoring of greylags tagged with GPS devices in Norway show that approximately 30–50% can stay in the Netherlands during the whole wintering season, whereas others migrate to France or Spain. These geese all return to their previous breeding sites, thus showing a high breeding site fidelity (Boos et al., 2012; M Boos, 2014, unpublished data). According to Ramo et al. (2012), an increasing number of greylag geese winter at higher latitude. A noticeable effect of climatic changes probably explains this increasing tendency for geese to winter more closely to their breeding grounds.

The European Atlantic flyway covers a vast area stretching from northern France to Spain and Portugal, with arrivals from Scandinavia, Poland, Denmark and Germany (Fouquet, Schricke & Fouque, 2009). The situation in France is particularly complicated, because noticeable fluxes of geese coming from northern or Central Europe are found not only along the Atlantic flyway but also in other areas located in central and south-eastern France. The departure areas of these birds have yet to be been fully determined, and the timing of migration can probably differ depending on the origin of the populations (Fouquet, 1991; Comolet-Tirman, 2009). Furthermore, the relative proportion of geese travelling to France and originating from different countries may change over time (Pistorius, Follestad & Taylor, 2006; Pistorius et al., 2007). However, data from neck-collared or ringed geese can be skewed by variations in the marking and resighting efforts of the countries involved (Nilsson, 2007; Nilsson et al., 2013), and this makes it difficult to fully define the composition of goose subpopulations migrating south from observational data alone.

Genetics have become a useful tool in the study of migration and wintering patterns. Recent studies on Anseriforms examined spatial structure along the flyways or in wintering zones, then compared it to genetic data in breeding areas. In the king eider (Somateria spectabilis), strong site fidelity to wintering areas and pair formation at wintering quarters indicated a population structure defined by wintering rather than nest-site philopatry (Pearce et al., 2004). However, genetic analyses of mtDNA and microsatellite alleles showed a lack of spatial genetic structure, suggesting the possible existence of flows with homogenized gene frequencies. In the mallard (Anas platyrhynchos), single nucleotide markers were used to investigate population structure on a continental scale throughout the northern hemisphere. This genetic analysis found a general panmixia, suggesting that mallards form a single large, interbreeding population (Kraus et al., 2013). The tufted duck (Aythya fuligula) shows high breeding site fidelity, but mtDNA and microsatellite markers revealed an extensive population admixture on the wintering ground (Liu, Keller & Heckel, 2012; Liu, Keller & Heckel, 2013). In the common pochard (Aythya ferina), genetic differentiation was observed among Eurasian breeding populations, but no evidence of genetic structure was detected for pochards sampled on European wintering grounds (Liu, Keller & Heckel, 2011).

Relatively few studies have investigated the genetic aspects of European geese of the genus Anser, and the subject has not been thoroughly investigated at all in the greylag goose. Studies by Ruokonen et al. (2004), Ruokonen, Aarvak & Madsen (2005) examined the genetic variability in two species of conservation concern, the lesser white-fronted goose (Anser erythropus) and the pink-footed goose (Anser brachyrhynchus), and investigated the phylogenetic relationship between seven Anser species (Ruokonen, Kvist & Lumme, 2000). A small amount of genetic differentiation between species has been observed in this genus (Ruokonen, Kvist & Lumme, 2000; Johnsen et al., 2010). Actually, mitochondrial DNA showed the presence of highly fragmented populations in two species of conservation concern, the lesser white-fronted goose (Anser erythropus, Ruokonen et al., 2004) and the pink-footed goose (Anser brachyrhynchus, Ruokonen, Aarvak & Madsen, 2005).

However, population genetics among species of geese has not been investigated to date. Here we used both mitochondrial DNA and microsatellites to study the characteristics of greylag geese from two breeding areas (the Netherlands and north-western Norway) and two wintering zones (northern France and south-western France). This study investigates to what extent populations are genetically differentiated. Genetic structure could have been increased by the fragmentation of breeding geese in separated areas, or on the contrary, a limited genetic structure could have been developed by (i) the widespread practice of amateur breeding and selling of geese (Hagemeijer & Blair, 1997; Wang et al., 2010), (ii) the recent increase in the size of several populations (Klok et al., 2010), and (iii) the habit of European geese to rest several times during their flight toward their winter quarters (Fouquet, Schricke & Fouque, 2009) at stopover sites where individuals from distant areas can admix and form pair bonds.

Knowledge of the genetic structure and diversity of greylag goose populations is a necessary scientific basis to manage this emblematic species (Lorenz, 1966) and decide on appropriate action for its conservation (Kampe-Persson, 2002) in the light of serious recent conflicts with agricultural and habitat protection interests in most north European countries (Klok et al., 2010).

Methods

Sample collection and DNA extraction

We analyzed feather samples from 174 greylag geese (Appendix S1) collected over the European Atlantic flyway (from two breeding grounds: in north and western Norway, and six staging grounds: in the Netherlands, northern France and south-western France; see Table 1 and Fig. 1). One additional individual was collected in the Republic of Kalmykia in an area associated with the eastern rubrirostris subspecies (Cramp, 1977). During the 2010/2011 and 2011/2012 (including 1–10 February) wintering seasons in France, goose feathers were obtained from greylag geese collected during the legal hunting period in natural areas by hunters collaborating with the study. Samples from the Netherlands were obtained on wild free-ranging geese collected in natural areas in the Zeeland region (near Rilland) by a local hunter before the 20th of September in 2011 and 2013, i.e., before the arrival of geese breeding in Norway or in Sweden (Nilsson, 2007; M Boos, pers. obs., 2014 based on GPS data). Samples from Norway were obtained from birds that were collected during the spring and summer legal hunting seasons, or from geese that were caught during the moulting period in 2010 and 2011 by A.F. for the Nordic Greylag Goose Project, which studies the ecology of the Norwegian breeding goose population (Nilsson, 2007). Feather calami were stored in ethanol at −20 °C, and total DNA was extracted using the commercial NucleoSpin® Tissue kit (Macherey–Nagel, Düren, Germany). After extraction, genomic DNA was stocked at −20 °C.

Table 1 Genetic variability of mtDNA CR1 D-Loop.

	Population	Season	N	Polymorphic sites	Hapl.	Private hapl.	h	π	K	D	FS	
South-western France	Charente-Maritime	Autumn- winter	10	16	7	1	0.933 (0.06)	0.01790 (0.00254)	5.156	−0.409	−0.664	
Gironde	Autumn- winter	20	12	7	0	0.821 (0.06)	0.01519 (0.0012)	4.374	1.052	1.190	
Landes	Autumn- winter	36	19	13	4	0.798 (0.06)	0.0161 (0.0013)	4.633	0.0376	−0.360	
North- France	Nord	Autumn- winter	17	10	7	0	0.833 (0.06)	0.01054 (0.0016)	3.025	0.014	0.433	
Oise	Autumn- winter	13	12	9	1	0.949 (0.042)	0.01309 (0.0020)	3.769	−0.103	−2.691*	
Northern Europe	Norway Finnmark	Late spring	11	6	3	1	0.564 (0.13)	0.00947 (0.002)	2.727	1.31175	3.038	
Norway Vega	Late spring	23	7	3	0	0.170 (0.10)	0.0021 (0.0016)	0.609	−2.147*	0,270	
Netherlands	Autumn	14	11	8	2	0.890 (0.006)	0.01244 (0.0013)	3.571	0.541	−1.112	
All Samples			144	20	23	–	0.823 (0.022)	0.01331 (0.0006)	3.819	−1.419	−4.515*	
Notes.

h haplotype diversity

π nucleotide diversity

D Tajima’s D

FS Fu’s FS

Standard errors are showed in brackets.

* P < 0.05.

Figure 1 Greylag goose distribution in Europe.

Main Anser anser flyways from breeding (red) to wintering (blue) areas (modified from IUCN, 2015). Pie charts indicate the proportion of different haplotypes (mtDNA) found in each sampled population. Colours are identical to those used in the haplotype network (Fig. 2), and haplotypes found in one area alone are the same colour.

Figure 2 Network and NJ tree.

(A) median-joining haplotype network. Areas of circles represent different sampled mtDNA haplotypes in proportion to their frequencies. Distances between haplotypes are proportional to the number of base differences. Colours match those utilized in Fig. 1, and haplotypes found in one area alone are the same colour. (B) neighbour-joining tree based on 280 bp of CR1. Sampled areas are labelled with abbreviations; NF, Northern France (Oise and Nord); SWF, South-Western France (Gironde, Charente Maritime and Landes); Nor, Norway; Neth, Netherlands. Numbers below branches indicate bootstrap values; only values above 50% are shown, most of the clades are supported by low bootstrap values.

Mitochondrial DNA sequencing

Partial mitochondrial control region (CR1 D-Loop 288 bp) was amplified in 144 of the 174 individuals (Appendix S1) using L180 (5′ tggttatgcatattcgtgcataga ′3) and H466 (5′ tttcacgtgaggagtacgactaat ′3) primers (Ruokonen, Kvist & Lumme, 2000). PCR amplifications were carried out in a Bio-Rad thermal cycler (Bio-Rad Laboratories Inc., Hercules, California, USA). PCR reaction was performed in a final volume of 25 µl containing 0.4 µl dNTPs (10 mM), 1 µl MgCl2 (25 mM), 0.3 µl of each primer (25 pmol/µl), 2.5 µl 10 × buffer, 0.4 µl Taq polymerase (5 unit/µl; QIAGEN), ddH2O and genomic DNA (20–100 ng/µl). The selected cycling profile included a 4 min preliminary denaturation cycle at 94 °C followed by 32 denaturation, annealing and extension cycles (30 s at 94 °C, 30 s at 58 °C and 30 s at 72 °C, respectively) before a final extension of 7 min. Negative controls were included for amplification procedures to detect contaminations.

The PCR product was purified using the EXO-SAP procedure with Exonuclease I (Exo; Fermentas, Burlington, Canada) and Shrimp Alkaline Phosphatase (SAP; Fermentas, Burlington, Canada). The purification cycle consisted of 30 min at 37 °C, then 15 min at 80 °C to deactivate the enzymes followed by a 10 min cooling-down step at 4 °C. DNA concentration was determined after electrophoresis in 1.8% agarose gels (TBE 1%) stained with ethidium bromide and visualized in a UV-trans illuminator Gel Doc XR (Bio-Rad Laboratories Inc., Hercules, California, USA) using the Molecular Imager ChemiDoc XRS System and Quantity One software (Bio-Rad Laboratories Inc., Hercules, California, USA).

Sequencing was carried out at Macrogen Laboratories (Amsterdam, The Netherlands) in an ABI 3730xl Analyzer (Applied Biosystems).

Raw electropherograms were checked visually using FinchTV (Geospiza Inc., Seattle, WA, USA; http://www.geospiza.com), and sequences were aligned with ClustalW algorithm in BioEdit 7.05 (Hall, 1999). The haplotype network was calculated in Network 4.6 (Fluxus Technology Ltd, Clare, Suffolk, England; fluxus-engineering.com) using the median joining procedure (MJ: Bandelt, Forster & Rohl, 1999). DnaSP version 5 (Librado & Rozas, 2009) was used to estimate mtDNA haplotype diversity (h), nucleotide diversity (π) and the mean number of pairwise differences (k) in the sampled areas. Demographic and/or spatial population expansion events were investigated using the mismatch distribution implemented in DnaSP v. 5. MEGA 5.0 (Tamura et al., 2011) was used to perform the neighbour-joining method (NJ: Saitou & Nei, 1987), clustering pairwise Tamura-Nei’s genetic distances between haplotypes (TN93: Tamura & Nei, 1993). Support for the internodes in the NJ tree was assessed by bootstrap percentages (BP: Felsenstein, 1988) after 1,000 resampling steps. One sequence of Anser anser anser (GenBank AF159962) from Finland and another of Anser anser rubrirostris (GenBank AF159963) from Slimbridge Wetland Center, England, were included as reference sequences in tree construction. A sequence of the lesser white-fronted goose (Anser erythropus, GenBank AY072580) and the bean goose (Anser fabalis, GenBank AB551534) were used as outgroups.

Maximum likelihood (ML) and maximum parsimony (MP) trees were obtained through the DNAML, CONSENSE, DNAPARS programmes in PHYLIP 3.67 (Felsenstein, 2005). Bootstrap values were based on 1,000 replicates, and the tree topologies were visualized with FigTree 1.3.1 (Rambaut, 2008). The best substitution model for molecular evolution was selected using the corrected Akaike Information Criterion (AICc, Burnham & Anderson, 2004) in jModelTest (Posada, 2008). Maximum likelihood bootstrap supports were estimated by performing 100 runs with 1,000 bootstrap replicates.

The partition of mtDNA diversity within and among the sampled geographical populations were investigated by running analyses of molecular variance (AMOVA, Excoffier, Smouse & Quattro, 1992) using Arlequin 3.3 (Excoffier & Lischer, 2010).

Microsatellite genotyping

A total of 169 of the 174 samples (Appendix S1) were genotyped by PCR amplification at 14 microsatellite loci (Ans02, Ans04, Ans07, Ans13, Ans17, Ans18, Ans21, Ans24, Ans25, Aalµ1b, Aph12b, Aph19b, Smo7b, Hhiµ1b) that had previously been isolated and tested in Anser anser (Weiß et al., 2008). We used PCR reactions, thermal profiles, fluorescent dye and multiplex sets, as indicated by Weiß et al. (2008). Microsatellite genotyping was performed on an ABI Prism 3100 Genetic Analyzer (Applied Biosystems) using the Macrogen Inc. GenScan service (Seoul, Korea). Negative controls were included for amplification procedures. Results were analysed in GeneMapper v. 4.0 (Applied Biosystems, Foster City, California).

Allele frequencies, standard diversity indices, observed heterozygosity (HO) and expected heterozygosity (HE) for each locus and population were calculated in GenAlex v. 6 (Peakall & Smouse, 2006).

We performed a factorial correspondence analysis (FCA) of individual multilocus scores in Genetix 4.05 (Belkhir et al., 2004) to describe genetic clusters.

Genepop 3.4 (Raymond & Rousset, 1995; Rousset, 2008) was used to calculate departures from the Hardy–Weinberg equilibrium (HWE) at each locus and within each population. Statistics were computed with Markov chain parameters at default settings.

We used Arlequin 3.5 (Excoffier & Lischer, 2010) to estimate the genetic variance within and between populations through a hierarchical Analysis of Molecular Variance (AMOVA; Excoffier, Smouse & Quattro, 1992).

The genetic structure of the sampled populations was computed using Bayesian clustering procedures in Structure v. 2.3 (Pritchard, Stephens & Donnelly, 2000; Falush, Stephens & Pritchard, 2003), without prior information about the origin and under an admixed model. Analyses were performed where K = 1–10 with 50 × 105 iterations following a burn-in period of 50 × 104 iterations; all simulations were independently replicated four times for each K. We explored the optimal value of K by plotting the average estimated LnP(D) (Ln probability of the data) and using ΔK statistics (Evanno, Regnaut & Goudet, 2005) calculated using Structure Harvester 0.6.93 (Earl & VonHoldt, 2012). Clumpp v. 1.1.2 (Jakobsson & Rosenberg, 2007) and Distruct v. 1.1 (Rosenberg, 2003) were used to align the cluster membership coefficients of the five Structure runs and display the results.

We investigated the presence of bottleneck events with Bottleneck v. 1.2.02 software (Cornuet & Luikart, 1996) for two models: the infinite alleles (IAM, Maruyama & Fuerst, 1985) and the two-phase model (TPM, Di Rienzo et al., 1994).

Migration rate was estimated using the Bayesian inference approach implemented in BayesAss 3.0.3 (Wilson & Rannala, 2003). We performed 10 runs of 9 × 106 iterations with a burn-in of 10%, and a sampling frequency of 200. Delta values were varied for all parameters, and resulted in acceptance rates between 40% and 60% of the total iterations (Wilson & Rannala, 2003).

Finally, isolation by distance was tested via Mantel tests with GenePop (Raymond & Rousset, 1995; Rousset, 2008; see also Legendre & Fortin, 2010); FST and geographic distance were compared using 1,000 random permutations. The geographic distance connecting samples was represented by Euclidean (linear geographic) distances computed in QG IS (QGIS Development Team, 2014).

Results

mtDNA

The mtDNA marker sequences (CR1 D-Loop 288 bp) showed 23 haplotypes defined by ten polymorphic sites and distributed in eight locations (Appendix S2). Among the 23 haplotypes found (GenBank accession numbers KT276333–KT276355), 14 haplotypes were shared by 2–47 individuals. According to the study areas, we found a total of nine private haplotypes, the majority of which occurred in the Landes (SW France) population (Table 1, Appendix S2). The diversity indices for mtDNA revealed moderate levels of genetic variation in the greylag goose in all sampled areas (Table 1). Haplotype diversity showed high values in all groups (range 0.798–0.949) except in breeding areas in Finnmark (0.564 ± 0.13 SD) and Vega, Norway (0.170 ± 0.10 SD).

The genetic distances recorded in Norway, northern France and the Netherlands were low (range 0.012–0.013). A slightly higher genetic distance was observed in south-west France in comparison to all other sampling sites (ranges 0.018–0.022), while the two breeding sites in Norway were genetically very close (Table 2).

Table 2 Tamura Nei genetic distance assessed by mtDNA.

Below the diagonal the genetic distance values and above the diagonal their standard errors.

	SW France - Landes	SW France - Gironde	N France - Nord	N France Oise	Netherlands	Norway - Finnmark	Norway - Vega	SW France Charente M	
SW France - Landes		0.004	0.004	0.004	0.004	0.005	0.006	0.005	
SW France - Gironde	0.016		0.004	0.004	0.004	0.004	0.005	0.005	
N France - Nord	0.015	0.014		0.004	0.004	0.004	0.004	0.005	
N France Oise	0.016	0.015	0.013		0.004	0.004	0.004	0.005	
Netherlands	0.015	0.014	0.013	0.013		0.004	0.004	0.005	
Norway - Finnmark	0.018	0.016	0.014	0.013	0.014		0.003	0.006	
Norway - Vega	0.017	0.014	0.012	0.012	0.012	0.008		0.006	
SW France Charente M	0.020	0.019	0.019	0.018	0.018	0.022	0.019		

The NJ tree shows that clades are composed of a wide variety of different geese from different areas. Individuals from each site were present in different branches (Fig. 2). Besides, none of the clades grouped together individuals originating from the same areas. About half the individuals were grouped together with the GenBank reference sequence relating to the anser subspecies, while the remainder were either grouped with the sequence relating to the rubrirostris subspecies or differed clearly from both subspecies. Very similar topologies were obtained from trees generated with other tree-building methods (MP and ML; not shown).

The haplotype median-joining network (Fig. 2) was concordant with the phylogenetic tree topology and did not reveal any geographic structures. The number of mutations separating the different haplotypes was low (max = 10).

Whilst 88.85% of the total genetic variance shown in hierarchical AMOVA was within populations, the remaining 11.15% occurred among populations. This indicates a small differentiation between the sampled areas.

Non-significant raggedness indices indicated a good fit to a model of population expansion in all sampled areas. Mismatch distribution results also suggested a population expansion in all areas except the Gironde region (P = 0.044) and Finnmark (P = 0.042) (Appendix S4). Fu’s FS value (Table 1) was only significantly negative for the Oise region, and was consistent with a demographic expansion for all other areas.

Microsatellites

Among the 15 microsatellites previously isolated by Weiß et al. (2008) Ans26 was shown to be monomorphic in all investigated individuals. The remaining 14 polymorphic microsatellite loci showed 2–12 different alleles per locus (n = 169 individuals; Table 3).

Table 3 Summary of genetic variation at 14 microsatellite loci.

	Population	N	Na	Ne	Ho	He	HWE (P)	
South-western France	Charente-Maritime	9	8.931 (0.071)	2.468 (0.340)	0.483 (0.082)	0.512 (0.051)	<0.001	
Gironde	24	4.286 (0.633)	2.575 (0.329)	0.462 (0.063)	0.504 (0.065)	<0.001	
Landes	45	4.643 (0.684)	2.625 (0.344)	0.484 (0.048)	0.542 (0.049)	<0.001	
North France	Nord	17	4.000 (0.584)	2.680 (0.332)	0.483 (0.072)	0.549 (0.056)	<0.001	
Oise	15	3.857 (0.573)	2.702 (0.403)	0.385 (0.061)	0.527 (0.063)	<0.001	
North Europe	Norway Finnmark	11	3.357 (0.372)	2.137 (0.266)	0.374 (0.072)	0.433 (0.065)	<0.01	
Norway Vega	34	4.571 (0.661)	2.485 (0.312)	0.476 (0.060)	0.529 (0.048)	<0.001	
Netherlands	14	3.929 (0.549)	2.352 (0.305)	0.447 (0.063)	0.493 (0.052)	<0.001	
All samples		169	4.027 (0.200)	2.503 (0.115)	0.449 (0.023)	0.511 (0.020)	<0.001	
Notes.

N number of individuals

Na No. of different alleles

Ne No. of effective alleles

Ho observed heterozygosity

He expected heterozygosity

Observed and expected heterozygosities were moderate, with similar values in each sampled population (Ho ranging from 0.374 to 0.484 and He from 0.433 to 0.549). Geese from the Landes wintering area exhibited the highest number of private alleles (n = 3, Appendix S3).

Genetic structure was visualized using factorial correspondence analysis (FCA) in each population (Fig. 3). The plot shows an absence of phylogeographic structure in the different investigated areas: individuals from different areas overlap, with the exclusion of four samples from Nord, one sample from Finnmark, one from Oise and one from Gironde.

Figure 3 Factorial correspondence analysis (FCA) of microsatellites data.

Outliers concern individuals from Nord (4 ind.), Oise, Gironde and Finnmark (1 ind.) populations.

Significant departures from HWE, due to heterozygote deficit and related to positive FIS values, were observed in all populations (Table 3, Appendix S3).

AMOVA analyses showed that 97.9% of the total genetic variance in geese was significantly distributed within populations (p < 0.001), while only 2.1% was distributed among populations. Overall fixation index F ST from AMOVA was 0.02105, indicating a low differentiation between areas.

Structure analyses, performed without the use of prior information on sample locations, showed a maximum ΔK at K = 4, while likelihood values reached a plateau at K = 7 (Fig. 4). Graphs show no evidence of phylogeographic structure across sampled populations, whatever the K value. With K = 4, only 23 individuals with individual qi values were each assigned to a single cluster: two individuals from Finnmark, three from Vega and one from Gironde were attributed to cluster 1; one individual from Netherland, two from Finnmark, six from Vega, one from Charente Maritime, two from Gironde and one from Landes were attributed to cluster 2; two individuals from Oise and two from Landes were assigned to cluster 3. All other birds had a highly mixed genotype. In the case of K = 7, five other individuals, one from Finnmark, Charente Maritime and Gironde and two from Vega, were assigned to the same cluster with qi > 0.90.

Figure 4 Structure analysis.

Estimated population structure in Greylag Goose sampled populations. Each vertical line represents one individual and each colour represents a single cluster.

Bottleneck events tested under IAM revealed a significant excess of heterozygotes (evidence of a recent bottleneck) in Nord, Landes and Oise populations (Wilcoxon sign-rank tests, all P < 0.05). Analysis under TPM only confirmed a recent bottleneck event for the Nord population (P < 0.05).

BayesAss detected a low migration rate among localities and a high proportion of local individuals (>68%, Table 4), suggesting that the flows among different areas were limited. Indeed, the analysis found a high proportion of local geese in six populations (>90%). In two cases, gene flow appears to be strongly asymmetrical, with many birds moving from Charente Maritime to the Netherlands (20.7% ± 3.79 SD) and from Oise to Gironde (20.2% ± 3.44 SD), but not in the opposite direction (1.5% and 1.1% respectively).

Table 4 Mean estimated number of migrants between breeding and wintering sites as calculated with BayesAss (standard deviations in parentheses).

Values on the diagonal (in bold) represent the estimated proportion of resident individuals in each population.

	Migration into	
Migration from	Netherlands	NF-Nord	NF-Oise	NO-Finnmark	NO-Vega	SWF-Charente Maritime	SWF-Gironde	SWF-Landes	
Netherlands	0.8904 (0.0338)	0.0167 (0.0160)	0.0151 (0.0147)	0.0152 (0.0144)	0.0160 (0.0154)	0.0152 (0.0146)	0.0157 (0.0151)	0.0157 (0.0148)	
NF- Nord	0.0145 (0.0139)	0.9009 (0.0316)	0.0135 (0.0130)	0.0140 (0.0135)	0.0144 (0.0138)	0.0133 (0.0129)	0.0149 (0.0143)	0.0145 (0.0141)	
NF- Oise	0.0142 (0.0135)	0.0460 (0.0238)	0.6820 (0.0146)	0.0144 (0.0139)	0.0138 (0.0132)	0.0136 (0.0132)	0.2023 (0.0344)	0.0137 (0.0133)	
NO- Finnmark	0.0191 (0.0183)	0.0212 (0.0198)	0.0174 (0.0165)	0.8619 (0.0394)	0.0215 (0.0210)	0.0177 (0.0164)	0.0219 (0.0211)	0.0194 (0.0184)	
NO-Vega	0.0085 (0.0081)	0.0078 (0.0078)	0.0078 (0.0078)	0.0081 (0.0079)	0.9439 (0.0192)	0.0079 (0.0077)	0.0081 (0.0079)	0.0081 (0.0077)	
SWF- Charente Maritime	0.2072 (0.0379)	0.0175 (0.0167)	0.0173 (0.0165)	0.0176 (0.0166)	0.0181 (0.0174)	0.6872 (0.0192)	0.0177 (0.0168)	0.0174 (0.0166)	
SWF- Gironde	0.0114 (0.0112)	0.0107 (0.0105)	0.0111 (0.0102)	0.0199 (0.0160)	0.0109 (0.0104)	0.0110 (0.0107)	0.9142 (0.0270)	0.0109 (0.0105)	
SWF- Landes	0.0069 (0.0069)	0.0068 (0.0067)	0.0063 (0.0063)	0.0067 (0.0066)	0.0065 (0.0064)	0.0063 (0.0063)	0.0068 (0.0067)	0.9538 (0.0164)	

The Mantel test calculated on geographic and genetic distances yielded a non-significant correlation coefficient (r = 0.107; P = 0.08), suggesting that there is no strong relationship between geographic and genetic distances.

Discussion

In this study we used a pool of 14 microsatellites isolated by Weiß et al. (2008) for greylag goose parentage in the long-established goose population at Konrad Lorenz Research Station, Grünau, Austria (Lorenz, 1966; Hirschenhauser, Möstl & Kotrschal, 1999). We found that these microsatellites can be successfully employed for geese sampled in a wide range of localities along the European Atlantic flyway. This is the first large scale study showing a moderate genetic variability of mtDNA and nuclear DNA in all French wintering areas and in the Netherlands, with slightly lower mtDNA variability in the Norwegian breeding sites. A moderate genetic variability in the greylag goose was already reported two decades ago by Blaakmeer (1995), and has been found in other species of geese (Anser erythropus: Ruokonen et al., 2004; Ruokonen et al., 2010; Anser brachyrhynchus: Ruokonen, Aarvak & Madsen, 2005). Low genetic variability also seems to be typical for other Anatidae species (Aythya ferina: Liu, Keller & Heckel, 2011; Aythya fuligula: Liu, Keller & Heckel, 2012). Interestingly, our results show that the genotypes deviated from Hardy–Weinberg expectation at eight loci, and in all study areas deviation was due to heterozygote deficiency. Additionally, the deficit of heterozygotes matched with positive FIS values. These results could be related to different factors such as population substructuring or recent population growth (Cornuet & Luikart, 1996).

Genetic distances between the different areas were low (range from 0.012 to 0.017) and the hierarchical AMOVA showed genetic variance to mainly occur within populations. These findings could be explained by a small differentiation between the sampled areas and a general admixture of greylag goose populations in our western European study region. However, it should be taken into account that genetic divergences in geese are characteristically very small, with the lowest interspecific divergence reported here for avian species (Ruokonen, Kvist & Lumme, 2000). The genetic tree shows that different branches include individuals from each sampling area. No single branch exclusively grouped together individuals originating from the same zone. Moreover, birds sampled in the western part of the breeding range, traditionally ascribed to the anser subspecies, were not separated from birds collected in the eastern part that were traditionally assigned to the rubrirostris subspecies (Kampe-Persson, 2002). Birds from Iceland, Scotland and coastal Norway have been sometimes separated as a race, sylvestris, classified in the anser group (Snow, Perrins & Cramp, 1998). Although Icelandic and Scottish birds were absent from this study, individuals from the Norwegian west coast did not appear to be clearly distinct from other European geese. Our present results slightly differ from the findings of Blaakmeer’s (1995) preliminary study, which reported genetic differences between breeders in two Dutch sites in comparison to breeding sites in south Sweden and Norway. However, Blaakmeer’s analyses show significant differences for only one of six minisatellites, in only two of the three Dutch areas studied.

Interestingly, the ANS19 sequence was recently found in the white Roman goose in Taiwan. This race is widely bred for commercial purposes, and has been found to originate from the European species (Anser anser, Wang et al., 2010). Our data confirm the presence of this sequence in Europe, particularly in the breeding population of the Norwegian west coast.

Haplotypes ANS02, ANS08, ANS11, ANS14 and ANS23 were only found during the winter in France, and were absent in Norway and the Netherlands: this result could indicate that some of the geese arriving in France came from areas we did not sample on the breeding grounds. Ring recoveries and resighting records indicate that these birds probably originated from northern Germany, Poland, Denmark and Sweden (Nilsson et al., 2013).

The haplotype network confirmed the tree configuration. There was no geographic pattern, and the number of mutations separating the different nodes was very low. This confirms the low genetic distance between our studied populations in the large north-western European population (as defined by Delany & Scott, 2006), and may reflect the rapid population expansion (Aris-Brosou & Excoffier, 1996).

Data obtained from nuclear DNA by microsatellites were in accordance with findings from mtDNA. As the mtDNA is uniparentally inherited whereas microsatellites are part of the biparentally inherited nuclear DNA, a difference between the two genomes would have indicated the presence of sex-biased dispersal (Fahey, Ricklefs & Dewoody, 2014). However, sex-biased dispersal seems to be unlikely in greylag geese for three reasons: the family unit remains together at least until autumn migration, the birds tend to pair before returning to the breeding grounds, and males and females have long-term pair bonds (Rohwer & Anderson, 1988; Doherty et al., 2002). Sex-biased dispersal in birds is probably not a species constant (Clarke, Saether & Roskaft, 1997). Within Anatidae in general, sex-biased dispersal was not detected in several species (Doherty et al., 2002; Mabry et al., 2013), while it was found in some species such as the white-fronted goose (Anser erythropus, Ruokonen et al., 2010), the common eider (Somateria mollissima, Paulus & Tiedemann, 2003), and the spectacled eider (Somateria fisheri, Scribner et al., 2001).

In our study of microsatellites, individuals from different geographic localities were found to be combined in the Factorial Correspondence Analysis representation. Bayesian structure analysis resulted in a best combination of four or seven groups, according to ΔK and LnP(D) methods respectively. As seen in our previous analyses, no geographic clustering was observed inside these Structure groups. Almost all individuals, with few exceptions, showed admixed genotype regardless of the number of groups considered in the analysis.

The high mixing of genotypes and the lack of geographic structure among our studied populations could be interpreted in the light of the data obtained through ringing activity and the extensive neck-banding programme carried out in Scandinavia from 1984 to 2004 (Nilsson, 2007; Voslamber, Knecht & Kleijn, 2010). Ring recoveries and visual observations showed that Scandinavian geese breeding in different zones can admix not only in the moulting areas (Nilsson, Kahlert & Persson, 2001) but also along the European Atlantic flyway, i.e., in the Netherlands (Voslamber, Knecht & Kleijn, 2010), France (Fouquet, Schricke & Fouque, 2009; Nilsson et al., 2013) and Spain (Ramo et al., 2012) where they can form pairbonds. Besides this Scandinavian data, the monitoring of collared and/or ringed individuals performed in other European areas showed the presence of birds in France originating from Germany, the Czech Republic and Poland. Populations that breed further east do not seem to reach France in winter (Kampe-Persson, 2010). From these data it appears that the greylag geese that cross France or winter there could result from a mixture of populations from different areas.

Our findings are somewhat unexpected if one assumes that the fragmentation of breeding populations into separate areas during the first part of the last century (Hagemeijer & Blair, 1997; Kampe-Persson, 2002), should have led to an increase in genetic structure. Moreover, birds breeding in the Netherlands have recently become highly sedentary (Fox et al., 2010), and this may also have contributed to the increase in genetic structure (Blaakmeer, 1995). However, a genetic panmixia could have been promoted by the widespread amateur breeding and selling of geese, and the recent increase and dispersal of several wild goose populations (Klok et al., 2010). In particular, geese with pink bills and legs, most probably rubrirostris subspecies, have been spreading in Europe over the last few decades; their natural flyway toward wintering areas crosses other European countries (from Russia to Hungary, the Balkan States and Italy) but does not reach France.

The breeding of geese is a widespread practice among amateurs, who can easily obtain both goslings and adults with a grey wild appearance (B and G Vaschetti, pers. comm., 2014). In some cases geese were released as part of assisted restoring projects and are now indistinguishable from the wild individuals (Kampe-Persson, 2010). Besides, birds with white plumage are common in breeding farms. In Asia, white geese are mostly descendants of the swan goose (Anser cygnoides). Even if descendants of Anser anser can also be found there, they are usually farmed in Europe (Wang et al., 2010). Although the two species can hybridize in captivity, hybrids can be detected through karyotype (Shahin, Ata & Shnaf, 2014) or genotype examinations (Sun et al., 2014). The contribution of escaped white form geese to the admixture observed in wild populations is probably low given the high assortative mating of wild greylag geese, their long-term monogamous pair bonds, female-bonded clan structure, long parent–offspring relationships, and elaborate patterns of mutual social support (Hirschenhauser et al., 2000; Kotrschal, Scheiber & Hirschenhauser, 2010).

Our findings on the greylag goose genetic admixture are similar to those reported in the snow goose (Chen caerulescens, Avise et al., 1992) and the barnacle goose (Branta leucopsis, Jonker et al., 2013). Despite the high rate of site philopatry seen in the snow goose, which has also shown a high increase in population over the last decades, mtDNA markers showed no clear distinctions between nesting populations across species range (Avise et al., 1992). The barnacle goose recently changed its migratory traditions, and new populations differing in migratory distance were observed. Genetic data showed an admixture between all populations, despite the assumed traditions of migration within areas and the presence of a newly established nonmigratory population in the Netherlands (Jonker et al., 2013). A lack of genetic structure in wintering areas was also found in four species of Anatidae, namely the common pochard, the mallard, the king eider and the tufted duck (Pearce et al., 2004; Liu, Keller & Heckel, 2011; Liu, Keller & Heckel, 2012; Kraus et al., 2013; Liu et al., 2013), and in the black-tailed godwit (Limosa limosa, Lopes et al., 2013). The mixing of breeding populations in wintering areas is believed to be a common phenomenon in birds, because the breeding ranges of most species are considerably larger than their wintering ranges (Winker & Graves, 2008). However, migratory populations vary in the degree to which individuals from distinct breeding localities mix on different sites. Therefore, to understand population demographics and genetic diversification, it is crucial to pinpoint which populations mix on breeding and wintering grounds (Chabot et al., 2012).

Our DNA-based estimates of migration during the wintering period indicated a low rate of exchange between our sampled areas. In five of eight areas the vast majority of individuals (86–95%) did not switch among the different zones, and a moderate exchange (about 20%) was only observed from Charente Maritime to the Netherlands and from Oise to Gironde. These results seem to support the hypothesis that the French wintering birds arrive from various areas, including zones that are not sampled here (i.e., Germany, Poland), while the contribution of the Norwegian population represents only a portion of the whole assemblage (Fouquet, Schricke & Fouque, 2009). This low exchange rate is also supported by evidence of great changes in spatial ecology recorded in 28 GPS tagged western European greylag geese, i.e., very small home ranges on wintering areas (<8.9 ± 2.5 km2) compared to 2–5 fold values obtained in staging areas during migratory and premigratory periods (M Boos, 2014, unpublished data).

Our data show evidence of genetic bottlenecks in just three groups under IAM, all located along the same flyway (Nord, Oise and Landes), and in a single case (Nord) under TPM. The discrepancy between IAM and TPM could be related to the different ability of the mutation models to detect bottleneck events. Empirical data suggest that TPM is the most appropriate model for microsatellite loci (Ellegren, 2000; Ellegren, 2004) while IAM results should be interpreted with caution (Cornuet & Luikart, 1996). We did not observe any sign of bottlenecks in the breeding populations: this indicates that greylag geese have never suffered any severe demographic reduction, even at the beginning of the past century when the number of breeding individuals was low in several European areas (Kampe-Persson, 2002).

Greylag goose populations are steadily increasing in north-western Europe (Kampe-Persson, 2002). The large number of geese in some areas is now in conflict with agricultural interests, since geese not only forage in natural environments but also forage on crop fields, and claims for a need to control the species are widespread (Klok et al., 2010). Our data suggest that the migratory geese harvested over France (about 20,000 geese/year, see Landry & Migot, 2000) show a relatively high diversity of origin. From this result it is difficult to conclude if there is a strong impact on a specific breeding population. Future studies could analyze other European breeding and wintering areas; this could clarify the status of the different populations and subspecies on the continent (the main Anser anser anser and A. a. rubrirostris, as well as the sylvestris forms from Iceland, Scotland and Norway), and help to build an effective international management strategy for this migratory species (Chabot et al., 2012).

Supplemental Information

Appendix S1 List of samples used in this study, with ID, population, sample provenance, collecting date, haplotype assignment and whether the sample was genotyped with microsatellites (STRs)

Click here for additional data file.

Appendix S2 Table of mtDNA haplotypes found in Anser anser individuals

Click here for additional data file.

Appendix S3 Summary of genetic variation at 14 microsatellite loci in sampled populations. Na, No. of different alleles; Ne, No. of effective alleles; Ho, observed heterozygosity; He, expected heterozygosity; F, fixation index; HWE, Hardy-Weinberg equilibrium

Click here for additional data file.

Appendix S4 Distributions of pairwise differences (mismatch distribution) among mtDNA haplotypes for overall dataset and each sampled areas

Click here for additional data file.

Our thanks to all the hunters who provided feather samples from France, to Wilmer Remijnse who provided feather samples from Netherlands, and to Paul Shimmings for his field assistance in Norway. We are grateful to Bruno and Gabriella Vaschetti who provided useful information on goose farming, the International Union for Conservation of Nature (IUCN) which provided an updated distribution map, and to Joanna Lignot-Munro for language editing.

Additional Information and Declarations

Competing Interests

Author Contributions

DNA Deposition

Arne Follestad is an employee of the Department of Terrestrial Ecology, Norsk Institutt for Naturfoskning and Mathieu Boos is an employee of Naturaconst@, Research Agency in Applied Ecology.

Irene Pellegrino conceived and designed the experiments, performed the experiments, analyzed the data, wrote the paper, prepared figures and/or tables, reviewed drafts of the paper.

Marco Cucco conceived and designed the experiments, analyzed the data, contributed reagents/materials/analysis tools, wrote the paper, prepared figures and/or tables, reviewed drafts of the paper.

Arne Follestad reviewed drafts of the paper, field work, collecting samples.

Mathieu Boos conceived and designed the experiments, contributed reagents/materials/analysis tools, wrote the paper, reviewed drafts of the paper.

The following information was supplied regarding the deposition of DNA sequences:

GenBank accession numbers: KT276333 to KT276355.

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
