# Peer review of "Lack of genetic structure in greylag goose (Anser anser) populations along the European Atlantic flyway"

_PeerJ, doi:10.7717/peerj.1161_

## Round 0.1 · original submission · Major Revisions

Dear authors
Thank you for submitting your manuscript to our journal. As you see our reviewers suggest a revision of your ms. If you are willing to do so, we would be happy to reconsider your revised manuscript.

Michael Wink

Reviewer 1 ·

Basic reporting

No Comments

Experimental design

No Comment

Validity of the findings

No Comment

Additional comments

The study ventures into analyzing the complex biogeographic relationships of the greylag goose, a species with highly dynamic populations across Europe and elsewhere. The species of this genus appear to be comparatively little differentiated among populations and studying the within-species variation is therefore inherently difficult. This was fully appreciated by the authors who nevertheless try to analyze the complex genetic relationships. Such an analysis was overdue and the efforts of the authors have to be appreciated. The obvious shortcoming of the study is the restriction to a few sampling sites, basically along the Atlantic coast (Fig. 1) that by no means reflects the complex pattern of breeding populations, regional seasonal movements and (obviously changing and dynamic) migratory patterns. The conclusions reached in this paper are therefore to be taken with caution.
Line 291: Given the low power of the Mantel-test (Legendre P, Fortin M J /2010/ Mol. Ecol. Resour. 10: 831-844) and the P-value of 0.08 and the general problem of type II errors, the statement “no relationship” is too strong (even when presented as an ‘indication’).
Figure 1 was largely taken from Comolet-Tirman 2009. The map there was compiled from Wikipedia-sources (sic!) and contains several geographically misplaced entries and especially gaps. So, for instance, sampling sites 1 and 2 would not lie in the breeding range of the species.

Reviewer 2 ·

Basic reporting

The authors made a great effort to sample Anser anser across western Europe focussing on Scandinavian breeding areas and French wintering sites. As typical for Anseriformes they failed to find a clear genetic substructure, at least with respect to geographic correlation. Since they used a large number of state-of-the-art bioinformatic tools to evaluate their DNA-sequence and microsatellite data, further data analysis does not promise to gain clearer results. The number and style of provided figures illustrates the (missing) data structure. Overall I consider this paper a valuable contribution to our knowledge.

Experimental design

Sampling, lab work, and data analysis were mostly following current standards (for exceptions see below).

Validity of the findings

The authors provide numerous potential explanations for the missing clear substructure of the goose population, but their final conclusions regarding the impact of hunting should not be drawn from the results of this study.

Additional comments

Please correct the following:
Consistently use upper- or lowercase for English names, provide scientific names with or without parentheses (ll. 57 vs. 74 and more often).
Use subscripts where appropriate (e.g. fst in l. 213 and more often).
References: In titles use uppercase only in the beginning and for names, in journal names for almost all "important" words, but not of, the and so on.
40 climatic changes or climate change
83 genetics
148 use proper acronym for dnasp
174 14
221 ten eight
261f four one one one
263 HWE not introduced before
272, 279 qi
291 A P value of .08 is obviously n.s., thus delete "n.s.". But is it marginally significant? Provide the actual correlation coefficient(s).
309 eight
310 fis
313 range?
318 interspecific
627 delete blanks from URI
722 mismatch
Fig. 2: Omit nodal support values below 70.
Consider the necessity of Fig. 3 in the main article.
Better explain what the outliers in Fig. 4 (could) mean.

---

## Round 0.2 · Minor Revisions

Dear authors

Thank you for submitting your revision. As you see, Reviewer 2 has identified some minor typos which you should correct in a final round of revision

Michael Wink

Reviewer 2 ·

Basic reporting

I see that the authors considered all comments by the reviewers. Only a few formal issues should be resolved prior to final acceptance (see below).

Experimental design

no changes in revision

Validity of the findings

no changes in revision

Additional comments

Epithet rubrirostris is misspelled rubirostris several times (both variants in Fig. 2!).
Northern, central and so on Europe are inconsistently capitalized.
l 114: remove gap from primer sequence
l 116: MgCl2: make 2 a subscript
ll 136, 139: inconsistent spelling of DnaSP/DNASP (other software programs in small caps)
l 180: should be K = 1–10
l 196: inconsistent spelling of QGis/QGIS
l 248: structure this time not in small caps
ll 301f: (1995) be moved immediately behind the author's name
l 312: northern Germany
ll 613f: Warbler. Wilson (no article)

---

## Round 0.3 · accepted · Accept

Dear authors,

Thank you for resubmitting your manuscript to our journal. We are happy to accept your ms now. Thank you for submitting your research results to our journal.